# Dynamics of the Emerging Genogroup of Infectious Bursal Disease Virus Infection in Broiler Farms in South Korea: A Nationwide Study

**DOI:** 10.3390/v14081604

**Published:** 2022-07-22

**Authors:** Tuyet Ngan Thai, Dae-Sung Yoo, Il Jang, Yong-Kuk Kwon, Hye-Ryoung Kim

**Affiliations:** 1Avian Disease Division, Animal and Plant Quarantine Agency, Gimcheon 39660, Gyeongsangbuk-do, Korea; ttn267@korea.kr (T.N.T.); kwonyk66@korea.kr (Y.-K.K.); 2Veterinary Epidemiology Division, Animal and Plant Quarantine Agency, Gimcheon 39660, Gyeongsangbuk-do, Korea; shanuar@korea.kr; 3Veterinary Drugs and Biologics Division, Animal and Plant Quarantine Agency, Gimcheon 39660, Gyeongsangbuk-do, Korea; moa305ho@korea.kr

**Keywords:** infectious bursal disease virus, antigenic variant, broilers, nation surveillance, random sampling

## Abstract

Infectious bursal disease (IBD), caused by IBD virus (IBDV), threatens the health of the poultry industry. Recently, a subtype of genogroup (G) 2 IBDV named G2d has brought a new threat to the poultry industry. To determine the current status of IBDV prevalence in South Korea, active IBDV surveillance on 167 randomly selected broiler farms in South Korea from August 2020 to July 2021 was conducted. The bursas of Fabricius from five chickens from each farm were independently pooled and screened for IBDV using virus-specific RT-PCR. As a result, 86 farms were found to be infected with the G2d variant, 13 farms with G2b, and 2 farms with G3. Current prevalence estimation of IBDV infection in South Korea was determined as 17.8% at the animal level using pooled sampling methods. G2d IBDV was predominant compared to other genogroups, with a potentially high-risk G2d infection area in southwestern South Korea. The impact of IBDV infection on poultry productivity or *Escherichia coli* infection susceptibility was also confirmed. A comparative pathogenicity test indicated that G2d IBDV caused severe and persistent damage to infected chickens compared with G2b. This study highlights the importance of implementation of regular surveillance programs and poses challenges for the comprehensive prevention of IBDV infections.

## 1. Introduction

Infectious bursal disease (IBD) is a highly contagious viral disease that is responsible for significant losses in the poultry industry worldwide [1]. The disease is caused by infectious bursal disease virus (IBDV) which attacks and destroys B-lymphocytes in the bursa of Fabricius (BF), leading to not only the surge of mortality in young chicks but also the increase in immunodeficient flocks vulnerable to secondary pathogen infection [2]. The IBDV genome consists of two segments, A and B, and the hypervariable (hv) VP2 region (amino acids 206–350) in segment A has been widely used as the primary determinant of the genetic evolution and antigenic variation of IBDV [3,4]. IBDV was first reported in Gumboro, Delaware in 1957 [5]. Since then, two serotypes (I and II) have been identified, of which only serotype I IBDV causes immunosuppressive disease in chickens.

Based on their antigenicity and pathogenicity, serotype I strains were originally categorized into four phenotypes: classic virulent (cv), antigenic variant (av), highly virulent (vv), and attenuated (at) [6]. However, owing to rapid genetic mutation in the hvVP2 region, a new classification of IBDVs with seven genogroups (G) has been proposed [7,8]. According to this improved scheme for IBDV genotype classification, the cv/atIBDV, avIBDV, and vvIBBV strains were identified as G1, G2, and G3, respectively. G2 was mostly prevalent in poultry farms in North America and Europe and further geographically distributed into three sub-lineages: G2a, G2b, and G2c [8,9]. Other genogroups G4 to G7 were later isolated and classified according to their geographic distribution or distinct molecular characterization from the original ones, indicating continuous mutation and recombination of the IBDV genome [8,10].

The first IBD outbreak in South Korea was reported in 1980, and since 1992, the prevalence of G3 IBDV has caused severe damage to chicken farms [11,12,13]. To prevent damage caused by G3 IBDV, the intermediate/intermediate plus vaccine is widely used in Korea. Nevertheless, a number of outbreaks of a novel type of G2 IBDV have been reported in several farms in South Korea [14]. This novel variant of IBDV was first found in China and was later classified as sub-lineage G2d, whose molecular characterization was different from that of the early variant IBDVs originally reported in America [9]. G2d IBDV has become widespread in China, mostly driven by evading the immune protection conferred by available vaccines [15,16,17]. More recently, G2d IBDV epidemics have also been reported in other Asian countries, including Japan [18] and Malaysia [19]. Under these circumstances, the lack of effective antigen-matched vaccines has contributed to the increasing prevalence of this new sub-genogroup in poultry holdings.

Under the potential circumstances that G2d IBDV may also dominate in South Korea, it is urgent to understand the prevalence rate of IBDV and evaluate its potential adverse impact on poultry production. Therefore, this study aimed to investigate the prevalence of IBDV in broiler farms across the nation from 2020 to 2021 and to identify high-risk areas for infection. Furthermore, the impact of IBDV infection on either productivity or secondary infection susceptibility and comparative pathogenicity between the two subgroups of G2 IBDV were also examined. This study provides a scientific basis for the development of an intervention strategy optimized for new variants of IBD outbreaks.

## 2. Materials and Methods

### 2.1. IBDV Surveillance

Nationwide surveillance of IBD in poultry farms across the nation was performed from August 2020 to July 2021. Two hundred meat-type chicken farms were randomly selected, taking samples from five chickens at slaughterhouses with ages ranging from 24 to 38 days that originated from each farm in nine municipalities. The number of farms for IBDV surveillance at slaughterhouses was determined based on the proportion of chickens slaughtered per abattoir in 2019 [20]. Details on the calculation of the designed sample size and the actual sample size can be found in Appendix A.

During the surveillance period, 33 meat-chicken farms were excluded from this study for the following reasons. First, 15 meat-chicken farms were excluded because 6 slaughterhouses were not in operation at the time of surveillance. Second, a sample from one farm could not be used because of poor storage conditions. Furthermore, among meat-type chicken farms, 17 farms raising minor chicken types such as White semi or Korean Native Chicken were excluded from the surveillance due to differences in age and weight at slaughter, and only commercial broiler farms such as Arbor Acre, Cobb, and Ross were investigated. As a result, a total of 167 farms were included in this study.

### 2.2. Animal Husbandry Status

The geographical location and number of birds at slaughter of the study farms were obtained from the livestock production safety management system. The production index (*PI*) at the study farm, which corresponds to productivity, was assessed using the following equation [21]. The *PI* for individual farms was obtained from companies operating slaughterhouses. This indicator was compiled from 108 broiler farms, except for 59 farms, where the company did not know or refused to provide information. The *PI* is defined as follows:$$Production\;index\;\left(PI\right)=\frac{mean\;weight\;at\;slaughter\left(kg\right)\times \left(number\;of\;live\;birds\;÷number\;of\;one-day\;chicks\right)}{days\;at\;slaughter\times \left(feed\;intake\;÷total\;body\;weight\right)}\times 100$$

### 2.3. Laboratory Diagnosis

For the detection of IBDV, BFs of five chickens from each farm were collected during necropsy examination and pooled as one sample. BF samples were homogenized in 10% (*w*/*v*) phosphate buffered saline (PBS) supplemented with 50 µg/mL gentamicin. The homogenate was then centrifuged at 3000× *g* for 10 min at 4 °C, and the supernatant was collected for subsequent IBDV detection and analysis. IBDV in these samples was detected using a virus-specific RT-PCR assay as described previously [8]. Briefly, viral RNA was extracted from bursa homogenates using TANBead nucleic acid extraction kits (Taiwan Advanced Nanotech Inc., Taoyuan City, Taiwan). RT-PCR was then conducted using Maxime RT-PCR Premix (iNtRON Biotechnology, Gyeonggi, Korea) under the following thermocycling conditions: 48 °C for 30 min; 94 °C for 5 min; 35 cycles of denaturation at 94 °C for 40 s, 57 °C for 40 s, and extension at 72 °C for 1 min; and final extension at 72 °C for 5 min. Primer05/01/95s 743-F (5′-GCCCAGAGTCTACACCAT-3′) and 1331-R (5′-ATGGCTCCTGGGTCAAATCG-3′) were used to amplify the 579-bp fragment of the hypervariable region of the VP2 gene (hvVP2). Amplified PCR products were electrophoresed on 1.5% agarose gels to detect an amplification of hvVP2 fragments.

Positive RT-PCR products were purified using a QIAquick PCR Purification Kit (QIAGEN, Hilden, Germany) and sequenced by Sanger sequencing (Bionics, Seoul, Korea). A 545-bp fragment of hvVP2 was assembled and aligned with reference IBDV strains using the CLC Workbench version 6.7.2 (CLC bio, Aarhus, Denmark). Sequences with 100% nucleotide similarities with those of IBDV live vaccines commercially used in South Korea were excluded from the prevalence study, and these farms were subsequently considered as negative for the virus infection. A phylogenetic tree was constructed using the neighbor-joining method and 1000 bootstrap replications using MEGA X software [22] and then displayed with iTOL (Interactive Tree Of Life) [23]. The partial VP2 gene sequences used in this study were deposited in the GenBank database under accession numbers ON715016-ON715115.

To diagnose colibacillosis caused by pathogenic *Escherichia coli* (*E. coli*), typical macroscopic lesions of colibacillosis were observed in organs such as the liver, air sac, and pericardium during necropsy. If such lesions were observed, pathogen isolation and identification were performed subsequently. Samples of the organs were cultured on a blood agar plate and MacConkey agar plate per sample, and the plates were incubated for 24 h at 37 °C. Bacterial species identification was performed using the VITEK-2 system (VITEK2 GN-card; bioMérieux, Marcy-l’Étoile, France) according to the manufacturer’s instructions [24].

### 2.4. Prevalence Estimation of IBDV

As pooled testing commenced with collection of BF samples from the 167 farms, we followed the protocol of animal prevalence estimation using pooled sampling methods, where 100% sensitivity and specificity for the test was assumed because the golden standard of virus detection by RT-PCR assay was applied to diagnose the infection. One to five BF samples were collected randomly from each broiler farm. Given these sample sizes, we assumed that the dilution did not affect the sensitivity of the test. Additionally, we assumed that all pooled samples were independent of each other. Under this assumption, the outcome of testing IBDV infection at farm *i* was denoted by *y_i_*, which was coded as one for positive and zero for negative from sample size *s_i_*, and followed a pooled Bernoulli distribution as follows [25]:$$\underset{i}{\prod }Bern\;(y_{i}|s_{i},\;p)={\left(1-p\right)}^{\underset{i}{\sum }s_{i}\left(1-y_{i}\right)}\underset{i}{\prod }{\left(1-{\left(1-p\right)}^{s_{i}}\right)}^{y_{i}}$$
where *p* is the animal prevalence, which is related to the probability of a positive result from the pooled sample, 1 − (1 − *p*)*s_i_*. Using these likelihood functions, we inferred animal prevalence using Bayesian inference with the Hamiltonian Markov-chain Monte Carlo method. The PoolTestR package version 0.1.2. was used based on the R software version 4.2.1. (Vienna, Austria, R core Team 2020).

### 2.5. Identification of Risk Area of IBDV Occurrence in Broiler Farms

Since the novel antigenic variant IBDV infection (i.e., G2d) has recently been reported, the geographical distribution of outbreaks could be limited to small areas, which is indicative of the initial phase of infectious disease outbreaks. Therefore, we conducted a spatial analysis to identify whether IBDV occurrence in broiler farms was clustered in a specific area. Three broiler farms were excluded from the cluster analysis because their geographical information was not available. For this analysis, the elliptic scan statistics were employed to calculate Bernoulli likelihood ratio (coded as one if the farm had a certain genogroup of IBDV infection (i.e., G2b, G2d, G3, or mixed infection of G2b and G2d) of interest, otherwise zero) for searching candidate clusters, limiting the maximum size to 50% of the total population of study at risk as default setting. The significance for the detection of clusters was determined by the *p*-value, which was obtained from a Monte Carlo simulation of 999 iterations based on the Bernoulli likelihood ratio function. The SaTScan software version 10.0.2 was used for spatial cluster analysis [26].

### 2.6. Adverse Effect of IBDV Infection on Broiler Production

IBDV is known to mainly attack B lymphocytes in the BF, leading to suppression of immunological function in chickens, increased vulnerability to secondary infection [27], and consequently declines in productivity, such as low feed conversion. Therefore, we conducted Bayesian multivariate mixed-effects linear regression to identify the association between IBDV infection and PI at the study farm.

Furthermore, Bayesian multivariate mixed-effect logistic regression was applied to examine the contribution of IBDV infection to *E. coli* infection. *E. coli* infection was diagnosed using samples collected from the same farms for IBDV testing.
$$E.coli\;infection_{ij}\;\sim \;B\left(\pi_{ij}\right),\;E\left(E.coli\;infection_{ij}\right)=\pi_{ij}$$
$$logit\left(\pi_{ij}\right)=\alpha +\beta_{1}*G2d_{ij}+\beta_{2}*G2b_{ij}+\beta_{3}*G3_{ij}+\beta_{4}*mixed\;infection_{ij}+random\;effect_{j}$$
$$random\;effect_{j}\;\sim \;N\left(\mu_{j},\;\sigma_{j}^{2}\right)$$

The mixed regression model assigned the farm location with respect to the spatial cluster to the random effect to account for different prevalence levels. In turn, *E. coli* infection *ij* corresponds to the outcome of *E. coli* infection at broiler farm *i* located inside or outside spatial cluster j for IBDV infection. The goodness of fit for the Bayesian logistic regression model was assessed using the area under the curve (AUC) of the receiver operating characteristic (ROC) curve for IBDV infection. R2jags package version 0.6–1 with R software was used for logistic regression using Markov-chain Monte Carlo (MCMC) sampling with 3 chains, 50,000 iterations, 4000 burn-in, and thin of 10. The Rubin-Gelman index (<1.05) was used to check the convergence of the MCMC chains [28].

### 2.7. Comparative Pathogenicity Study of IBDV Strains Belonging to Different G2 Sub-Genogroups

IBDV isolates used in this pathogenicity study, G2b strain 19D38 and G2d strain 20D39, were isolated from a passive surveillance performed in 2019 and 2020, respectively [14]. Isolation and titration of IBDV strains were performed in 10-day-old specific pathogen-free (SPF) chicken embryonated eggs via the chorioallantoic membrane (CAM) route, as previously described [29]. Viruses were harvested from CAMs and examined to exclude contamination by other avian pathogens using molecular assays. Determining the median embryo effective dose (EID_50_) was performed by virus titration following the method described by Reed and Muench [30].

All experimental animal procedures were approved and supervised by the Institutional Animal Care and Use Committee of the Animal and Plant Quarantine Agency of Korea. Sixty-three one-day-old SPF chickens were purchased from a local company (Namduk SPF, Gyeonggi, Korea) and divided into three groups (*n* = 21 birds/group) housed in separate isolators provided with feed and water ad libitum. At 21 days of age, chickens in group 1 and 2 were challenged through oculo-nasal route with a dose of 10^5^ EID_50_/0.1mL of IBDV G2b strain 19D38 and G2d strain 20D39, respectively. Chickens in the control group were used as negative controls and mock-inoculated with phosphate-buffered saline solution. At 1, 3, 5, 7, and 14 days post-challenge (dpc), three birds from each group were euthanized, weighed, and examined for macroscopic lesions post-mortem. BFs were weighed to calculate the bursa:body weight (B/BW) ratio and bursal body weight index (BBIX). The mean BBIX value was calculated along with the standard deviation [BBIX = (B/BW ratio in the infected group)/(B/BW ratio in the control group)]. BF with a BBIX value less than 0.7 was considered as atrophy [31]. Viral loads in BFs and viral shedding in cloacal (CL) swabs were examined by fluorescent quantitative real-time RT-PCR (qRT-PCR) using IBDV-specific primers and probes. The same titrated stock of challenge virus was used to establish the standard curve for viral load calculation, and data were indicated in EID_50_/0.1 mL. BF samples collected at 1, 3, 5, and 7 dpc were stored in 10% neutral formalin for histopathological examination. Briefly, paraffin-embedded tissues were routinely processed, and 4-µm-thick sections were prepared and stained with hematoxylin and eosin. Microscopic lesions in BFs were scored in order of severity from 0 for normal BFs to +4 for severely infected birds, according to a previous study [32]. The mean lesion scores (MLSs) were calculated for each group. At 7 and 14 dpc, blood was collected from all chickens, and their sera were tested for anti-IBDV antibodies using the commercial ELISA IBD-XR Ab test kit (IDEXX Laboratories, Maine, USA) according to the manufacturer’s instructions.

All statistical analyses were performed using GraphPad Prism version 8.4.3 (GraphPad Software, San Diego, CA, USA). The statistical significance of the differences among the different groups was determined using two-way analysis of variance. Differences were considered significant at *p* < 0.05 (*), *p* < 0.01 (**) and *p* < 0.0001 (****).

## 3. Results

### 3.1. IBDV Detection from Surveillance and Its Genogroup

Out of 167 broiler farms in this study, 103 farms were confirmed to have infection with field IBDV, which accounted for 61.67% of tested farms. There were 33 farms not infected with IBDV in which BF samples were negative for IBDV-specific RT-PCR assay. Samples collected from 31 farms were detected with commercial live vaccine strains, which were also considered as not infected with field IBDV. According to the phylogenetic analysis based on hypervariable regions of the VP2 gene, 86 farms were infected with the G2d variant, 13 farms were infected with the G2b variant, 2 farms were infected with G3, and 1 farm named 21RI008 was found to be concurrently infected with both G2b and G2d variants (Figure 1).

### 3.2. Prevalence Estimation of IBDV Infection

Table 1 provides the prevalence estimates of IBDV infection in broilers in South Korea, based on randomly sampled pooled tests. At the animal level, approximately 17.8% of broilers in South Korea were infected with various genogroups of IBDV. Specifically, there was evident discrepancy in G2-specific prevalence where the G2d viruses (mean = 0.140, 95% credible intervals [95% CrI] = 0.113 − 0.168) predominantly infected broilers than other genogroups of IBDV.

### 3.3. Identification of Risk Area of IBDV Occurrence in Broiler Farms

As shown in Figure 2, one spatial cluster for G2d IBDV outbreaks in broiler farms located in the southwestern part of the country was identified. Inside the cluster, 77.4% of total G2d variant incidence was confirmed, resulting in relative risk of 2.02. In contrast, other genogroups of IBDV did not display a geographically clustered outbreak in broiler farms.

### 3.4. Association of Productivity of Broiler and Secondary Infection to E. coli

Table 2 summarizes the regression coefficient estimates of IBDV infection for the productivity index estimated using Bayesian multiple mixed-effect linear regression. According to the results, the broiler farms infected with the G2d IBDV showed a decline in their PI, but not a significant drop compared to non-infected farms. G3 viral infection decreased the PI of broiler farms by the highest number, but no significance was reported. In addition, G2b virus-infected broiler farms showed a lower decline in PI than other types of IBDV infections.

The association between IBDV infection with *E. coli* infection is shown in Table 3. The broiler farms with G2d IBDV infection reported 2.64-times higher risk of *E. coli* infection (95% Crl = 1.16–6.24) compared to IBDV non-infected premises. However, the other genogroups (i.e., G2b or G3) of IBDV infection did not have a significant impact on *E. coli* infection, with a broader CrI containing an odds ratio of 1.

### 3.5. Comparative Pathogenicity Study of IBDV Strains Belonging to Different G2 Subgroups

Considering the high prevalence rate of G2 IBDV in South Korea and the possible adverse effects of IBDV infection, an in vivo pathogenicity trial was performed in SPF chickens. During the study, neither clinical signs nor mortality were observed in any of the experimental groups. Atrophy of the bursa was observed from 3 dpc in all virus-challenged groups and lasted until the end of the experiment. No post-mortem lesions were observed in the control birds at any time point. BBIX values for virus-challenged groups were <0.7 from 3 dpc and continued to decrease to 0.25 and 0.27 at 14 dpc in the 19D38 and 20D39 virus-challenged groups, respectively, confirming the results of the post-mortem examination (Figure 3).

Histopathologically, typical lesions of BFs infected with IBDV, including loss of epithelium, lymphocyte depletion, and lymphoid follicle diffusion, were found in the G2 IBDV-infected BFs, although no lesions were found in the BFs of the control group (Figure 4). In particular, according to the mean lesion scoring, the G2d 20D39 strain caused more severe microscopic lesions in the BFs than the G2b 19D38 strain at 5 and 7 dpc (Figure 5).

The viral loads in the BFs and CL swabs were determined using qRT-PCR analysis. Viral RNA was not detected in the control group. The virus started to replicate in BFs at 1 dpc in both virus-challenged groups, and the viral loads peaked at 3 dpc, then gradually declined over the course of the experiment. There was no significant difference in viral replication in BFs infected with the different IBDV G2 subgroups. Viral shedding also peaked at 3 dpc for the virus infection groups, and viruses were detected in cloacal swabs until 7 dpc (Figure 6).

Antibody responses in chickens infected with IBDV strains are shown in Figure 7. No IBDV-specific antibody was obtained for the control group at all time points. A positive antibody response to both variant IBDV strains was observed from 7 dpc onwards, and an increasing antibody titer was observed at 14 dpc. Remarkably, chickens infected with G2d strain 20D39 had significantly higher antibody titers than those infected with G2b strain 19D38 at 14dpc (*p* < 0.05).

## 4. Discussion

In an effort to keep track of the current IBDV status in South Korea, we attempted to conduct active surveillance on 167 commercial broiler farms randomly selected across the country. This diagnostic approach could be considered reliable for monitoring the current prevalence of the disease, since the detection of IBDV is not traditionally focused on birds with clinical signs of IBD or any other diseases. Remarkably, 61.67% of tested farms were confirmed to have IBDV infections. Attempts to determine prevalence estimates at the animal level, based on random sampling pooled test, revealed that 17.8% of broilers in South Korea had been infected with IBDV. Moreover, three genogroups, G2b, G2d, and G3, were found to be present in poultry farms across the nation. Notably, G2b continues to circulate in South Korea since two G2b strains were isolated in 2019 [12]. Specifically, there was an evident discrepancy in genogroup-specific prevalence, where the G2d virus predominantly infected broilers compared to other IBDV genogroups. These findings reflect the characteristics of the current IBDV epidemic in South Korea, where there has been an outbreak of IBDV infection in domestic broiler farms with the G2d virus becoming the dominant genogroup.

IBDV G2d infection in broiler farms, the most prevalent genogroup of infection, showed a geographically clustered pattern. This clustered area is known to have the highest density of chicken farms in South Korea and has experienced several highly infectious disease outbreaks, such as highly pathogenic avian influenza [33]. A similar finding was reported in China, where G2d IBDV was also prevalent in the major poultry-raising regions of this country [17]. This implies that the novel sub-genogroup of IBDV is likely to harbor a vulnerable area and facilitate inter-farm transmission at close distances during the course of the epidemic. Our findings highlight the urgent need to formulate an intensive biosecurity standard for highly vulnerable areas to prevent the spread of infectious diseases.

Concerted efforts to control domestic G3 IBDV outbreaks for nearly 30 years, mainly including widespread vaccination and improved poultry feeding management, might have contributed to effectively controlling the G3 epidemic in South Korea. Although vaccination information on the farm could not be fully obtained in this surveillance due to a lack of or wrong information from farm owners, this situation was confirmed by our surveillance results with only 2 out of 167 randomly examined broiler farms across the country found to be infected with this genogroup. However, a recent study showed that current commercial vaccines against G3 IBDV were unable to provide complete protection to chickens against G2 variant viruses [34]. This finding correlated with the fact that G2 variants were isolated from poultry flocks immunized with commercial intermediate vaccines [14,19,35,36]. Several attempts to developing vaccine candidates against G2d variants has been done [37,38]. Moreover, some commercial recombinant vaccines have been reported to induce protection against a wide range of IBDV strains, including G2 variants [39]. The fact that genogroup 2 IBDV, especially G2d, is currently becoming a dominant genogroup in South Korea poses an urgent need for a proper update on the current vaccination program.

Antigenic variant IBDV infection-induced immunodeficiency in chickens has been well reported to make the flock susceptible to secondary infections [6,40]. In this study, we found that broiler farms with G2d virus infection reported a higher risk of *E. coli* infection compared to IBDV non-infected farms. This finding was consistent with a recent study that found that high mortality observed in 30 day-old chickens in low-performance broiler farms infected with G2d IBDV in Japan could be due to secondary infections such as *E. coli*, probably one week after IBDV infection, which was at the time of immunosuppression in these broilers [18].

Due to high resistance to most disinfectants and environmental factors, IBDV continues to be present in poultry houses and tends to reappear in subsequent chicken flocks. Consequently, acute or chronic IBD in the flock reduces the efficiency of broiler production and net profitability. A 5-year study of the incidence and economic impact of various IBDVs on broiler production in Canada found that IBDV prevalence was probably attributed to substantial economic losses of approximately 3.9 million kg per year to the poultry industry [41]. Although the data were not significant, our findings confirmed the possible adverse impact of IBDV infection on poultry productivity, with broiler farms infected with the G2d variant showing a decline in their productivity index compared to non-infected farms. Some data in our study did not show a significant difference. This may have been due to the exclusion of more than 30 farms during the survey (Figure 1). Furthermore, G2 IBDV infection has been considered a subclinical infection; therefore, it may be difficult to see the remarkable adverse association of G2 IBDV with productivity with short-term surveillance.

Since the G2d variant was first identified in China and has been widespread throughout Asian countries, several studies on its pathogenicity have been performed [16,19,42,43]. In this study, we compared the pathogenicity of two IBDV strains belonging to different IBDV G2 sub-genogroups: IBDV G2b strain 19D38 and G2d strain 20D39. Neither strain caused any obvious gross clinical symptoms or mortality, which correlated with previous studies considering variant IBDV infection as a subclinical infection. Moreover, the two isolates showed relatively similar effects on bursal atrophy, viral shedding, and replication efficacy in the bursa. However, at a histological level, G2d isolate 20D39 caused more severe lesions in BFs than G2b isolate 19D38. A protective antibody response occurred from 7dpc for both virus-challenged groups despite the potentially immunosuppressive effects on the bursa. Antibody titers against IBDV in chickens in the G2b-challenged group were slightly higher than those of G2d group at 7dpc, whereas at 14dpc the antibody response was significantly reversed. A similar pattern of antibody response was observed in a recent comparative pathogenicity study of a variant of very virulent IBDV [43]. The results may suggest that there was a weaker humoral immune response to G2b strain than the G2d strain at the acute phase, but a reverse incidence was observed at the chronic phase. Furthermore, in a study evaluating the antibody response against IBDV vaccines with different level of pathogenicity, an intermediate plus vaccine induced higher antibody response than an intermediate vaccine [44]. These findings may indicate that higher pathogenicity levels of IBDV G2d than G2b may correlate with the antibody response against those sub-lineages. To our knowledge, this is the first report to evaluate the pathogenicity of two sub-lineages of G2 IBDV.

## 5. Conclusions

We found that IBDV was highly prevalent in South Korea, with G2d the dominant genogroup, particularly in dense areas with a high density of broiler farms, and had an adverse impact on productivity efficiency as well as susceptibility to secondary infection. Pathogenicity studies revealed that both G2 sub-genogroups induced bursal atrophy, leading to an immunosuppressive effect, with G2d causing more severe histopathological lesions. Furthermore, many studies have provided adequate evidence that current commercial G1-derived vaccines against G3 IBDV do not effectively protect poultry against variants of IBDV, underscoring the urgency for an updated vaccine program. Without pharmaceutical interventions available for the emergence of novel viruses, this study suggests the development of a control strategy that includes intensive risk management in clustered areas and underscores the importance of the implementation of regular surveillance programs. Therefore, it is necessary to develop systematic intervention strategies to minimize nationwide disease transmission.

## Figures and Tables

**Figure 1 viruses-14-01604-f001:**
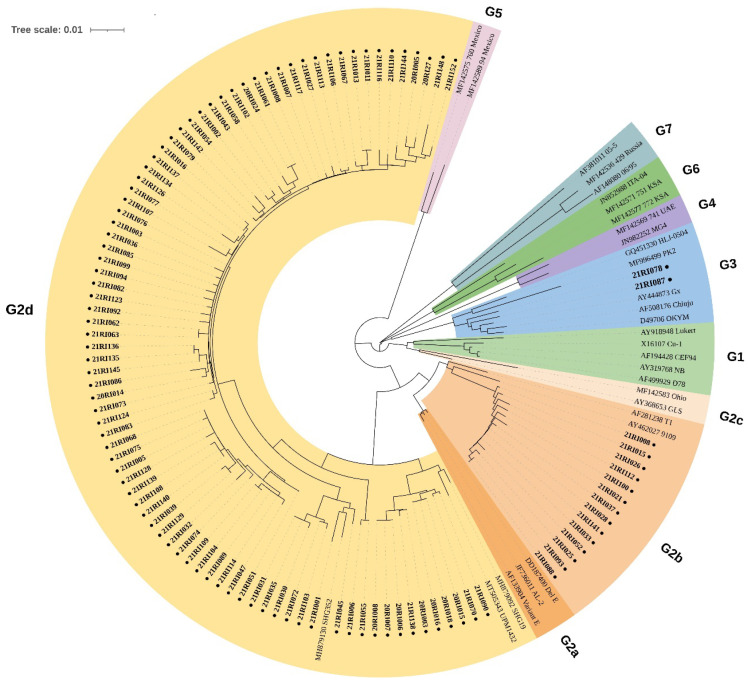
Phylogenetic analysis of the nucleotide sequences of the hypervariable region of the VP2 gene. Broiler farms infected with IBDVs found in this study were indicated as bold with a solid circle mark.

**Figure 2 viruses-14-01604-f002:**
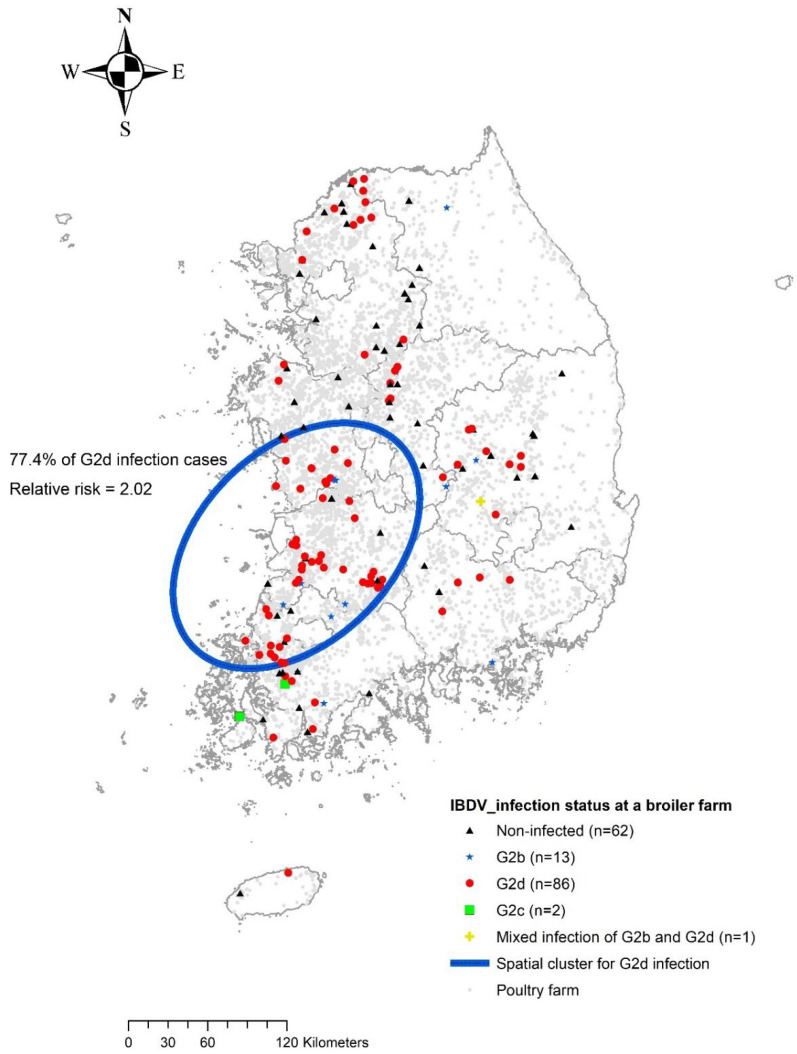
Geographical distribution of IBDV-infected premises and the spatial cluster for G2d variant of IBDV infection denoted by a blue circle.

**Figure 3 viruses-14-01604-f003:**
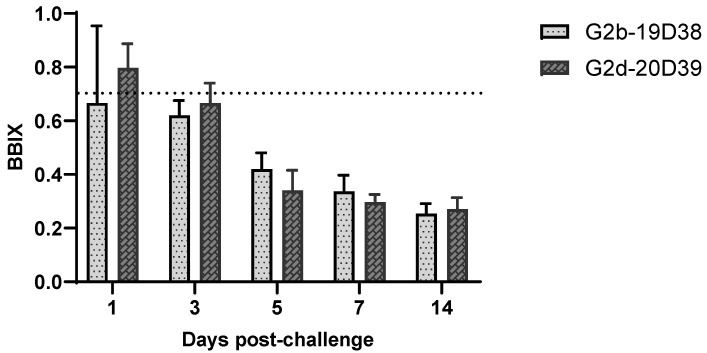
The bursa:body weight index (BBIX) of chickens challenged with IBDV strains 19D38 and 20D39. The dotted horizontal line indicates threshold of bursal atrophy. Error bars indicate standard deviation.

**Figure 4 viruses-14-01604-f004:**
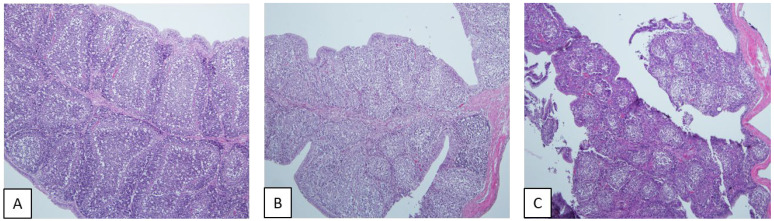
Histopathology of the bursas of Fabricius of specific pathogen-free chickens at 7 days post-challenge (× 100 magnification). (**A**) Control birds: normal lymphoid follicles of the bursa. (**B**) 19D38 virus-infected birds: moderate atrophy in bursal lymphoid follicle is diffuse. (**C**) 20D39 virus-infected birds: severe atrophy, bursal lymphoid follicle diffusion and exfoliation of epithelium.

**Figure 5 viruses-14-01604-f005:**
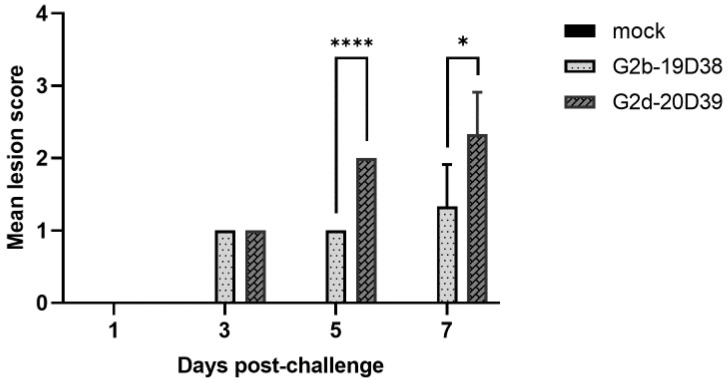
Mean lesion scores in bursas of Fabricius of chickens challenged with variant IBDV strains. Error bars indicate standard deviation. *, *p* < 0.05; ****, *p* < 0.001.

**Figure 6 viruses-14-01604-f006:**
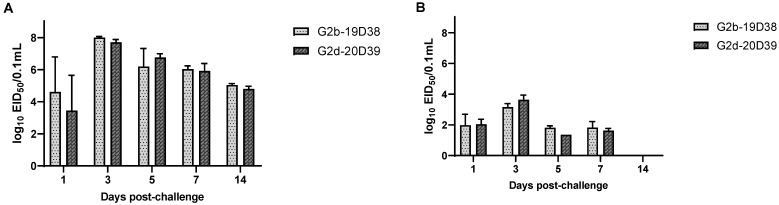
Viral loads of antigenic variant strains in (**A**) bursas of Fabricius and (**B**) cloacal swabs from chickens as determined by qRT-PCR analysis. Error bars indicate standard deviation.

**Figure 7 viruses-14-01604-f007:**
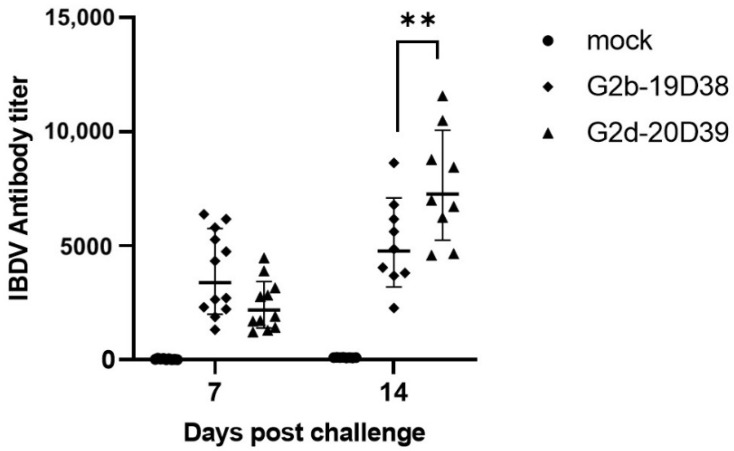
Serum antibody titers to IBDV of the control group and challenged groups. Each data point represents individual antibody titers, and the horizontal bar indicates geometric mean titers. **, *p* < 0.01.

**Table 1 viruses-14-01604-t001:** The prevalence estimation of infectious bursal disease of broilers in South Korea.

Category	Mean	95% Credible Intervals	
		Lower	Upper
IBDV			
G2d infection	0.140	0.113	0.168
G2b infection	0.017	0.087	0.026
All IBDV infection ^a^	0.178	0.148	0.209

^a^ All IBDV infection consisted of G2d, G2b, G3 and mixed infection (G2b and G2d).

**Table 2 viruses-14-01604-t002:** Association of IBDV infection with productivity index at broiler farms (*n* = 108) by Bayesian linear mixed effect regression.

Variables	Coefficient Estimates ^a^		
	Mean	95% Credible Interval	
		Lower	Upper
G2d infection	−8.651	−28.243	10.941
G2b infection	−5.340	−38.189	27.510
G3 infection	−30.550	−93.171	32.067
No. of chicken slaughtered	0.081	−0.212	0.374

^a^ The coefficient estimates of G2d, G2b and G2c infection for productivity index (PI) represents the change of PI value when the corresponding genogroup of infection occurred against the non-infection.

**Table 3 viruses-14-01604-t003:** Odds ratio of infectious bursal disease infection for the presence of *Escherichia coli* in commercial broiler farms (*n* = 167) by Bayesian mixed-effect logistic regression.

IBDV Infection (Reference = Not Infection)	Odds Ratio		
	Mean	95% Credible Interval	
		Lower	Upper
G2d infection	2.64	1.16	6.24
G2b infection	3.05	0.74	11.90
G3 infection	5.09	0.13	20.60
Mixed infection (G2b and G2d)	0.02	0.00	17.20

## Data Availability

The hvVP2 sequences of IBDV strains in this study are available in the GenBank database under accession numbers: ON715016-ON715115.

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
