# Peer review of "Dynamics of the Emerging Genogroup of Infectious Bursal Disease Virus Infection in Broiler Farms in South Korea: A Nationwide Study"

_viruses, 2022, doi:10.3390/v14081604_

Round 1
Reviewer 1 Report
In this manuscript Thia et al present the prevalence of IBDV genogroups in South Korea. Here are some of my suggestions/comments:
Major deficiency of this work is lack of IBDV vaccines used in each of the sites that the samples were collected. Without that data it is difficult to speculate why certain genogroups and sublineages are prevalent in certain areas. The authors acknowledge this short coming in the discussion section (L355).
Figure 3 and corresponding results and discussion is misleading. I doubt that G2d is clustered in southwest region of South Korea. This circle could as well be drawn in any other regions for example you could have it in the north region. This clustering is more a factor of poultry density. I recommend that the map shows the poultry density in shades and reanalyze if there is any clustering.
In the conclusion, L410 are you saying that currently in South Korea, they use IBDV vaccines belong to G3? If this is true, can you provide what is the percent use of various vaccines that are currently used?
Other points:
Figure 1 can be omitted because it is described in detail L83-90
Please label all figures with lineage not just strain. Fore example instead of saying 19D38, it should give the sub lineage
L240. Rephrase, it gives wrong meaning that way it is written
Figure 3. Spelling of Chinese variant
It maybe more appropriate to use words that are more specific to birds like flock prevalence than herd.
Author Response
Response to Reviewer 1 Comments
Point 1: Major deficiency of this work is lack of IBDV vaccines used in each of the sites that the samples were collected. Without that data it is difficult to speculate why certain genogroups and sublineages are prevalent in certain areas. The authors acknowledge this short coming in the discussion section (L355).
Response 1: We admit that major drawback of our work is lack of vaccination information. It is inevitable due to large number of farms being sampled and all of them were randomly chosen for the active surveillance. Despite of this limitation, we believe that our study is adequate and convincing enough to show current situation of IBDV prevalence in South Korea with G2d becoming dominant genogroup.
Point 2: Figure 3 and corresponding results and discussion is misleading. I doubt that G2d is clustered in southwest region of South Korea. This circle could as well be drawn in any other regions for example you could have it in the north region. This clustering is more a factor of poultry density. I recommend that the map shows the poultry density in shades and reanalyze if there is any clustering.
Response 2: As the reviewer commented, we have added the poultry farm distribution as a background to the map of the spatial cluster of IBDV infection in broiler farms (Figure 2). The cluster analysis using binomial scan statistics searches the numerous candidates of the spatial cluster with different shapes and sizes across the study regions, in this case, the entire territory in South Korea. If other regions have significantly more cases of G2d infection, it would be certainly detected. As the reviewer mentioned, density is a well-known contributor to infectious disease infection, mainly due to higher contact rates between the farms. In this sense, the location of the spatial cluster suggests the poultry farm density is positively associated with G2d outbreaks in broiler farms in South Korea, which we already discussed in discussion section.
Point 3: In the conclusion, L410 are you saying that currently in South Korea, they use IBDV vaccines belong to G3? If this is true, can you provide what is the percent use of various vaccines that are currently used?
Response 3: Currently, there is no G3-derived vaccine in use in South Korea. Most commercially available conventional live IBDV vaccines are based on classical virulent strains (later classified as G1), which can protect chickens from G3 IBDV infection. We admit that our writing may cause misunderstanding to readers. Therefore, we have amended the sentence to avoid this misunderstanding.
Other points:
- Figure 1 can be omitted because it is described in detail L83-90.
- We have deleted Figure 1 and rearranged other figures in order.
- Please label all figures with lineage not just strain. For example, instead of saying 19D38, it should give the sub lineage.
- We agreed with reviewer’s comment. In all figures about pathogenicity test, we have added the sub-lineage after the virus strain name (19D38 was changed to G2b-19D38, 20D39 was changed to G2d-20D39).
- Rephrase, it gives wrong meaning that way it is written.
- According to the reviewer’s comment, we have rephrased the sentence.
- Figure 3. Spelling of Chinese variant.
- We have corrected this misspelling.
- It may be more appropriate to use words that are more specific to birds like flock prevalence than herd.
- We have corrected this in abstract and discussion section.
Reviewer 2 Report
Peer-reviewed work presents the results of surveillance of Infectious bursal disease virus (IBDV) on 167 randomly selected broiler farms in South Korea in 2020-2021. More than 800 samples were screened for IBDV using virus-specific RT-PCR and subtype of virus was determined. A qualified statistical analysis was carried out. It was shown that 61.67% of tested farms have infection with field IBDV. Subtype G2d of genogroup2 was predominant compared to other genogroups. Impact of IBDV infections on productivity of broiler was estamated. Comparative pathogenicity trial in vivo of G2b and G2d IBDV strains was performed. Histopathology, viral loads and antibody responses was determined
Remarks
Line 204: “Virus titration was performed by determining the median embryo effective dose (EID50) 204 following the method described by Reed and Muench .”
I think well be better: “Determining the median embryo effective dose (EID50) was performed by virus titration following the method described by Reed and Muench”
Line 320-322: “Remarkably, chickens infected with G2d strain 20D39 had significantly higher antibody titers than those infected with G2b strain 19D38 at 14dpc (P < 0.05).”
The data on Figure 8 are more complicated. On day 7 post challenge antibody titers is higher in G2b group, whereas on day 14 antibody titers is higher in G2d group. Authors must comment that discrepancy
Author Response
Response to Reviewer 2 Comments
Point 1: Line 204: “Virus titration was performed by determining the median embryo effective dose (EID50) 204 following the method described by Reed and Muench.”
I think well be better: “Determining the median embryo effective dose (EID50) was performed by virus titration following the method described by Reed and Muench”.
Response 1: The sentence was amended according to the reviewer’s suggestion.
Point 2: Line 320-322: “Remarkably, chickens infected with G2d strain 20D39 had significantly higher antibody titers than those infected with G2b strain 19D38 at 14dpc (P < 0.05).”
The data on Figure 8 are more complicated. On day 7 post challenge antibody titers is higher in G2b group, whereas on day 14 antibody titers is higher in G2d group. Authors must comment that discrepancy.
Response 2: Following the reviewer’s comment, we have amended the discussion about this discrepancy in discussion section.